# Denoising of the Fiber Bragg Grating Deformation Spectrum Signal Using Variational Mode Decomposition Combined with Wavelet Thresholding

**Weifang Zhang [1], Meng Zhang [1], Yan Zhao [2], Bo Jin [2] and Wei Dai [1,***

[1] School of Reliability and Systems Engineering, Beihang University, 37 Xueyuan Rd., Haidian Dist., Beijing 100191, China; 08590@buaa.edu.cn (W.Z.); zhangmeng123@buaa.edu.cn (M.Z.)

[2] School of Energy and Power Engineering, Beihang University, 37 Xueyuan Rd., Haidian Dist., Beijing 100191, China; zy_buaa@buaa.edu.cn (Y.Z.); by1504121@buaa.edu.cn (B.J.)

* Correspondence: dw@buaa.edu.cn; Tel.: +86-138-1058-4286



**Featured Application: The main purpose of this study is to present a precise denoising algorithm to denoise the fiber Bragg grating (FBG) deformation spectra signal. The method combines variational mode decomposition and wavelet thresholding. It can be applied in signal processing in the structural health monitoring field.**

**Abstract:** Damage detection using an FBG sensor is a critical process for an assessment of any inspection technology classified as structural health monitoring (SHM). FBG signals containing noise in experiments are developed to detect flaws. In this paper, we propose a novel signal denoising method that combines variational mode decomposition (VMD) and changed thresholding wavelets to denoise experimental and mixed signals. VMD is a recently introduced adaptive signal decomposition algorithm. Compared with traditional empirical mode decomposition (EMD), and it is well founded theoretically and more robust to noise samples. First, input signals were broken down into a given number of $K$ band-limited intrinsic mode functions (BLIMFs) by VMD. For the purpose of avoiding the impact of overbinning or underbinning on VMD denoising, the mixed signals, which were obtained by adding different signal/noise ratio (SNR) noises to the experimental signals, were designed to select the best decomposition number $K$ and data-fidelity constraint parameter $\alpha$. After that, the realistic experimental signals were processed using four denoising algorithms to evaluate denoising performance. The results show that, upon adding additional noisy signals and realistic signals, the proposed algorithm delivers excellent performance over the EMD-based denoising method and discrete wavelet transform filtering.

**Keywords:** fiber Bragg grating; variational mode decomposition; discrete wavelet transform; signal processing

---

## 1. Introduction

Fiber Bragg gratings (FBGs) have attracted more and more attention due to their small size, high resolution, multiplexing capability, immunity to electromagnetic fields, and other interesting features. Recently, FBG sensors have been considered as promising in structural health monitoring (SHM) [1]. When the central wavelength of the reflected light shifts with the introduced structure strain/stress, the FBG sensor performs as an optical strain gage for strain/stress measurements [2]. Furthermore, FBG sensors have shown great potential for monitoring applications in aluminum fatigue

crack by analyzing the deformation spectrum signal with crack propagation [3]. However, the truthful signal data contains various noises caused by the environment, personal operation, and other reasons.

In this paper, to identify and abstract valuable information from the initial signal, combined with background noise, especially under a high-noise condition, we propose wavelet thresholding in a variational mode decomposition (VMD) domain signal denoising method, called the VMD-DWT denoising algorithm. The VMD was first proposed by Dragomiretskiy and Zosso in Reference [4]. It is an ensemble, non-recursive variation decomposition mode, the basic mode being named band-limited intrinsic mode functions (BLIMFs) $u_k$, where the modes are extracted concurrently and their central frequencies $w_k$ are estimated online using the alternate direction method. The research shows that the frequency of BLIMFs is more compact than that of intrinsic mode functions (IMFs), and the VMD outperforms the empirical mode decomposition (EMD) in signal–noise separation and robustness based on some experimental data. Thus, the VMD can be used to remove the noise from non-linear and non-stationary signal. In Reference [5], the VMD was used to denoise the vibration signals caused by rotor-to-stator rubbing, and the analysis results show its superiority over empirical wavelet transform (EWT) [6], ensemble empirical mode decomposition method (EEMD) [7], and EMD [8]. In Reference [9], VMD was successfully used to denoise a biomedical image, and in Reference [10], a typical EGG signal was denoised using only VMD. Moreover, Zhang et al. [11] studied chatter detection in a milling process based on the entropy of the basic modes decomposed using VMD and wavelet packet decomposition (WPD). Jiang et al. [12] demonstrated a new coarse-to-fine decomposing strategy of the VMD that can detect the weak repetitive transients of heavy noise signals at a high quality. Recently, researchers have verified the basic decomposition modes of VMD related to spectrum energy. The decomposition level $K$ decides the energy distribution in each basic mode [13]. However, in the literature, there are few studies on the definition method of decomposition level number $K$. In Reference [6], the decomposition level of VMD coincides with that of the EMD decomposition number. Moreover, in Reference [13], VMD combined with detrended fluctuation analysis (DFA) is proposed to denoise noisy signals corrupted by white Gaussian noise (WGN), and the criterion based on DFA is designed to select the number $K$. Gao et al. [14] applied simulated Acoustic emission (AE)signals to its wave packages adaptively, and the number of wave packages is seen as the value $K$. However, the reasonability of the operability of the decomposing strategy of VMD for the experimental FBG signals are seldom discussed because it is difficult to establish the basic components of the signals. Obviously, these predefined methods are disproportionate, and the selection method of the best decomposition number $K$ is worth investigating.

FBG signals detected under fatigue loading are a non-stationary time series. The degree of noise interference in damage detection processing is changed with the diverse external load environment. It is clear that wavelet denoising methods are widely used in signal processing, and a discrete wavelet transform (DWT) can effectively remove noise and obtain the detailed information of the signal [15,16]. In the case of VMD, the thresholding operation should be properly adapted to be consistent with the energy characteristics of the decomposition basic modes. Inspired by improved wavelet thresholding [17], in this paper, we propose a novel denoising method that combines variational mode decomposition with the changed thresholding discrete wavelet transform, called the VMD-DWT algorithm, to process experimental signals.

Several EMD denoising algorithms have been developed, such as EMD-soft [18], EMD-DFA [19], and EMD-changed thresholding wavelet [20]. EMD [8] is a recursive procedure to analyze non-linear and non-stationary multicomponent signals by deposing them into several amplitude- and frequency-modulated (AM/FM) zero-mean principle mode signals, which are without predefined basis vectors. However, EMD decomposition highly depends on the extremal point findings method. A lack of a mathematical theory and the problem of recursive shifting and mode mixing in the extraction process of IMF by EMD have inspired such solutions as empirical wavelets [21] and recursive variational decomposition [4]. Nevertheless, these attempts have only partially addressed the drawbacks of EMD. Recently, VMD, due to its advantages in noise resistance, the balance of

the backward error, and a narrow-band definition of meaningful modes, has been proposed as an alternative to EMD for dividing multi-component signals into different modes based on a clear variational model [22]. VMD combined with a discrete wavelet thresholding algorithm for denoising mixed and real experiment signals under various working conditions presents a superior effect compared with traditional denoising based on EMD methods.

The proposed approach has two key advantages over the previous studies. First, the novel changed thresholding VMD-DWT algorithm can be effectivity applied to denoise the FBG signals that are obtained from the fatigue crack propagation detection experiment. Once the crack is initiated, the amplitude fluctuation of the noise spectral signal measured from the experiment would be obviously changed [3]. Therefore, a VMD combined with the changed thresholding wavelet method is proposed to satisfy the future precise denoising demand of the damage signals. Second, although the VMD or the DWT denoising algorithm has been studied in other signals, such as the EEG [11] and the vibration signals [6], the combined denoising algorithm is seldom adapted in fatigue crack experiment FBG signals. Moreover, the energy distribution in each basic mode has been considered in wavelet thresholding. Additionality, compared with other decomposition parameter definition methods, in this paper, it focuses on the physical characteristic of the signal and the real experiment environment. Through analyzing the denoising results of different methods in mixed and real signals, the proposed denoising method shows good performance in FBG signal denoising processing in the SHM field.

The remainder of the paper is organized as follows. We present the experimental design in Sections 2 and 3 provides a brief description of VMD and the major concepts of DWT. Meanwhile, it also discusses the possibility of adapting the wavelet thresholding principle in VMD decomposition. Consequently, novel changed thresholding VMD-DWT strategies are presented. Section 4 explores two cases. In Case 1, four denoising algorithms are applied to denoise the mixed signals. The processing results show the effectiveness of the proposed method with respect to FBG signals, and the best decomposition number $K$ and data-fidelity constraint parameter $\alpha$ were obtained. In Case 2, real experimental signals used to evaluate the novel denoising techniques are illustrated. This section also includes a discussion of the results. In Section 5, the final conclusions are drawn.

## 2. Experimental Design

The purpose of FBG sensor damage detection is to analyze the deformation spectrum and reflectivity intensity. Their bandwidths change with crack propagation. Spectrum characteristic analysis results can be used to local the damage and quantify the severity. In this study, an aluminum alloy structural component equipped with FBG sensors was designed to achieve crack damage detection.

### 2.1. Specimens: Material and Geometry

Aircraft grade 2024-T3 aluminum alloy coupons 2.0 mm in thickness were manufactured at the AECC Beijing Institute of Aeronautical Materials (Beijing, China). The specimens with dimensions of $300 \times 100 \times 2$ mm, a straight 10 mm hole in the center of the plate, and a 2 mm through-thickness pre-crack was processed by electric discharge machining (EDM) in the two sides of the hole to develop a fatigue crack. The detailed geometry of the specimen is shown in Figure 1. All specimens have the same geometry and were made of the same materials. The yield strength of the materials was 360 MPa, the ultimate strength was 490 MPa, the Poisson's ration was 33, and the Young's modulus was 72,000 MPa. Results from Reference [23] indicate that the pre-crack around the hole has a larger stress concentration and therefore has the most potential to develop the fatigue damage.

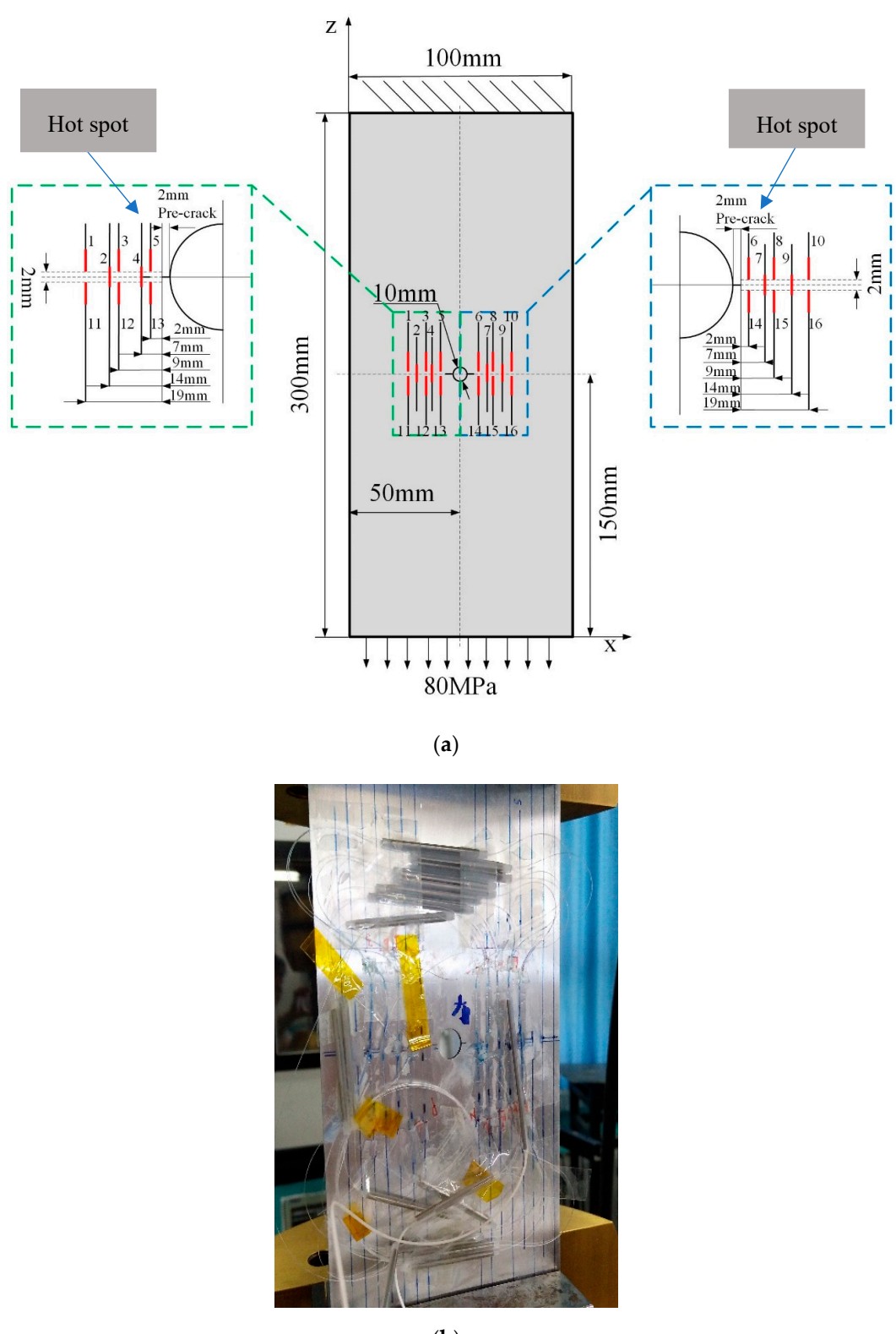

**Figure 1.** (**a**) Schematic of the aluminum specimen and the FBG sensor layout. (**b**) Metallic homogeneous coupons bonded with the FBG sensors under the fatigue experiment test.

## 2.2. Sensor Layout Design

The FBG sensor layout design is critical for damage detection. The sensors were bonded on the surface of the plate to sense the damage condition. According to the existing experimental data from

our pre-research testing and the finite element analysis at different crack lengths [23], the crack usually initiates at the pre-crack position, grows on both sides of the hole, and then finally breaks the specimen. Thus, the central hole region was marked as the target area. Sixteen FBG sensors bonded by the liquid cyanoacrylate adhesive were uniformly placed on both sides of the hole. The adhesive Young's modulus of the FBG sensor was 1.7 MPa, and the length was 10.1 mm. Furthermore, the deformation spectrum introduced by the crack damage was caused by the axial strain profile. Considering that, the FBG sensors were placed at the terminal of the crack tip. The detail sensor placement layout is shown in Figure 1a, where red lines represent the FBG sensors near the target region.

### 2.3. Experimental Setup

The experimental platform for fatigue hole-edge crack damage detection contained three major parts: an optical sensing and data acquisition system, a fatigue crack measurement system, and a fatigue load-cycling system, which are shown in Figure 2. FBG sensors (FSSR5025) manufactured by the Changcheng Institute of Metrology and Measurement (Beijing, China), were used to monitor the crack propagation behavior. The optical demodulator system (SM125, Micro Optics Inc., Danbury, CT, USA) was adapted to record the reflection spectrum at various crack lengths from 1 to 30 mm, and the measurement accuracy was 1 $\mu\varepsilon$. Regions vulnerable to fatigue crack damage were monitored with a traveling optical microscope and a charge coupled device (CCD) camera during loading, and the discovered crack lengths were defined as the true crack lengths. The fatigue cycling load testing was conducted using a hydraulic Mechanical Testing & Simulation (MTS) machine at room temperature, and the constant amplitude tensile loading was applied to the bottom of the plate, with a fixed top boundary. The amplitude was 80 MPa, the stress ratio was 0.1, and the load frequency was 5 Hz. In addition, the fatigue testing experiment was paused for data acquisition, and the processing was repeated twice to eliminate the operation error during each of the pauses. The overall experimental setup is shown in Figure 2. The experiment signal data were investigated to evaluate the effect of the de-noising algorithm and to monitor the size and location of the crack damage.

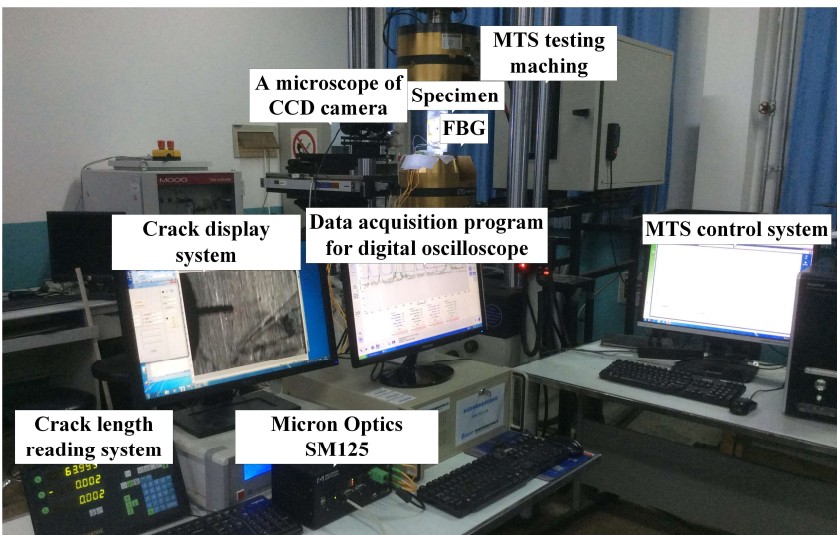

**Figure 2.** Experimental setup for the FBG sensor damage detection system.

### 2.4. Experimental Results

Fatigue cracks are naturally generated during fatigue testing. In fatigue testing, the healthy and damaged signals are collected for damage monitoring. Figure 3 shows a plot of the reflectivity spectrum versus the crack length for the specimen. In the following figure, external loads lead to uniform or non-uniform strain field distributions along the sensor grating. When the strain field is uniform, the FBG reflection spectrum only shifts in the amount proportional to the applied strain

(assuming isothermal conditions) in Figure 3a. However, when the strain is non-uniform, the spectrum shifts and distorts at the same time. Moreover, the spectrum broadens and shows multiple peaks in Figure 3b.

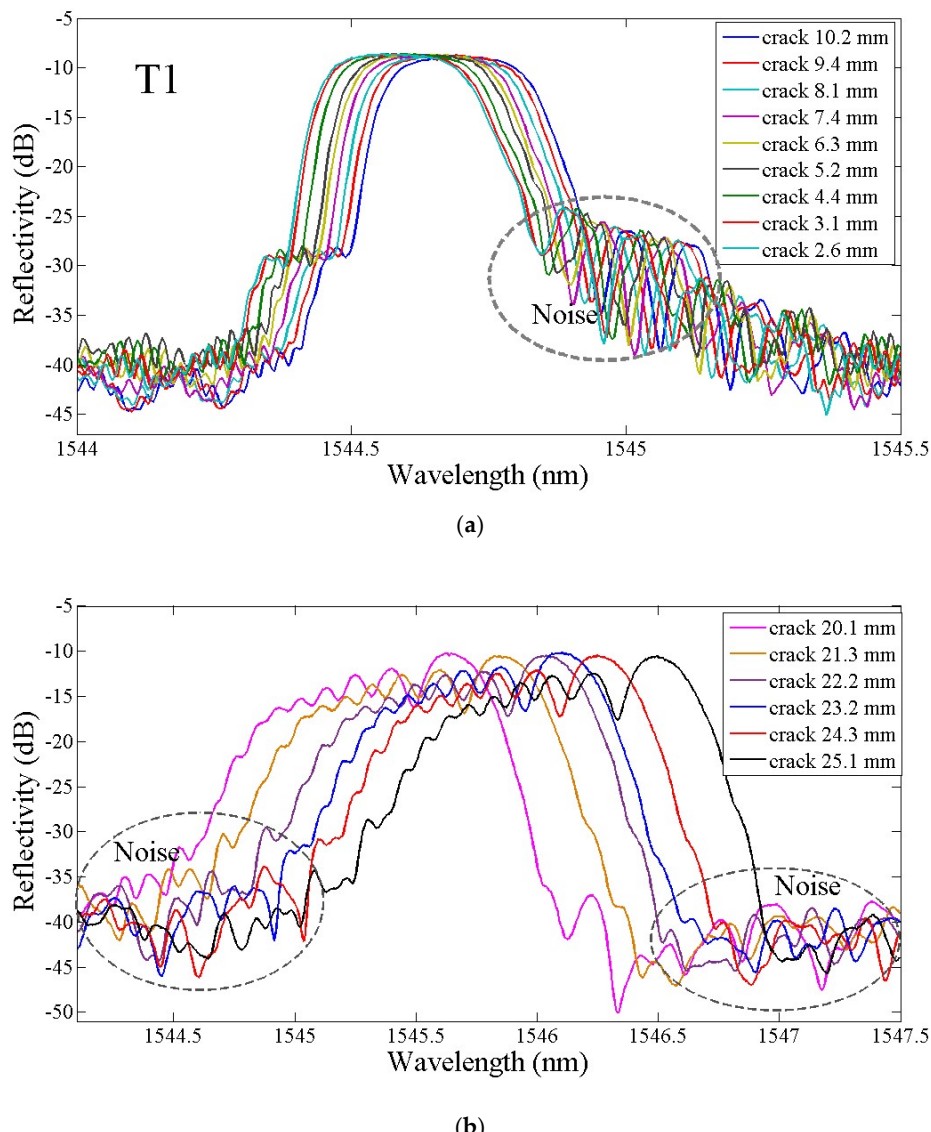

**Figure 3.** The reflection spectrum at different crack lengths obtained from the experiment. (**a**) The ideal signals, and (**b**) the heavy noise signals.

According to previous research [3], when the results of the crack damage monitoring experiment are ideal, the damage characteristics extracted from the deformation spectrum can be used to detect crack damage. However, it can be seen in Figure 3b that partial experiment results obtain a substantial amount of noise. The FBG sensing system is influenced by electrical items and the external environment, and the noise is expected to be included in the reflectivity spectrum. The noisy irregular leap points are magnified in the figure below with black cycles. Additionally, the noisy experiment signal affects crack damage detection precision, so a precision denoising algorithm is required to estimate the initial free noise signal from the experimental signal before the feature detection operation.

### 3. The Principle of VMD-DWT-Based Denoising

During the late 1990s, Huang introduced the EMD algorithm [8]. It is widely used to recursively decompose a signal into different unknown and separate spectral bands. However, the properties of the EMD method includes a sensitivity to sampling noise and a lack of mathematical theory, drawbacks that limit its application in signal de-noising. Though some attempts have been made to address these issues, such as empirical wavelets and the recursive VMD mode, they continue to persist. In 2014, Konstantin [22] introduced VMD, an entirely non-recursive variational decomposition mode. In his research, VMD was described as different from the existing decomposition in that it can extract the mode concurrently from the input signal online. Moreover, the mode is an ensemble of modes, respecting central frequencies $w_k$. The basic modes are obtained by applying the VMD algorithm to the input signals, called band-limited intrinsic mode functions (BLIMFs) $u_k$ with respect to the sub-energy of the signal. As the decomposition number is predefined, the decomposition level decides the energy distribution in each basic mode. The authors in Reference [4] show the high-order modes $u_k$ representing fast oscillations and the low-order modes $u_k$ representing slow oscillations. The noise is almost in the high-frequency domain, so the experiment signal should be separated into several high-order modes. For the purpose of enhancing denoising performance, inspired by the translation invariant wavelet, VMD-based denoising techniques were developed. The wavelet-changed thresholding principle was employed in the decomposition basic modes. The VMD-changed thresholding algorithm is described in this section.

### 3.1. Brief Description of VMD

The goal of VMD is to decompose the input signal into sub-signal modes, $u_k$, which have specific sparsity properties. The basic modes $K$ are most related to the center frequencies $w_k$. A detailed description of VMD can be found in Reference [4]. In order to assess the modes $u_k$ and the frequencies $w_k$, the resulting constrained birational problem, described as follows, needs to be solved using VMD:

$$\min_{\{u_k\},\{w_k\}} \left\{ \sum_k \left\| \partial_t \left[ \left( \delta(t) + \frac{j}{\pi t} \right) * u_k(t) e^{-jw_k t} \right] \right\|_2^2 \right\} \tag{1}$$

$$\text{Subject to } \sum_k u_k = f \tag{2}$$

in which the $u_k$ and $w_k$ are shorthand notations of the mode and its center frequency. The $\delta$ denotes the Dirac distribution, $\partial_t$ denotes the partial differential, and $*$ indicates convolution.

The reconstruction constraint can be addressed by making use of both a quadratic penalty term and Lagrangian multipliers $\lambda$. The argument Lagrangian L is expressed as follows:

$$
\begin{aligned}
L(\{u_k\},\{w_k\},\lambda) = \quad & \alpha \sum_{k=1}^{K} \left\| \partial_t \left[ \left( \delta(t) + \frac{j}{\pi t} \right) * u_k(t) \right] e^{-jw_k t} \right\|_2^2 \\
& + \left\| x(t) - \sum_{k=1}^{K} u_k(t) \right\|_2^2 + \left\langle \lambda(t), x(t) - \sum_{k=1}^{K} u_k(t) \right\rangle
\end{aligned}
\tag{3}
$$

where $\alpha$ represents the balance parameter in the data fidelity constraint and corresponds to the bandwidth of the mode. The solution to the problem is the alternate direction method of multipliers (ADMM). The Winner filtering in the Fourier domain is adopted to update the modes $u_k$. According to Reference [4], the embedded Winner filtering in the VMD algorithm makes the modes more robust to sampling and noise. The sub-mode $u_k$ in the time domain and the filtered analytic signal transformed by inverse Fourier can then be defined as follows:

$$\hat{u}_k^{n+1}(w) = \frac{\hat{f}(w) - \sum_{i \neq k} \hat{u}_i(w) + \frac{\hat{\lambda}(w)}{2}}{1 + 2\alpha(w - w_k)^2} \tag{4}$$

$$\hat{u}_k(t) = R\{ifft(\hat{u}_k(w))\} \tag{5}$$

in which the $\hat{f}(w)$ shows the Fourier transform of the signal $f(t)$. The $R$ denotes the real analysis signal part, and the $ifft(.)$ expresses the inverse Fourier transform of the signal.

The optimal $w_k$ by Fourier domain is given as follows:

$$w_k^{n+1} = \frac{\int_0^\infty w|\hat{u_k}(w)|^2 dw}{\int_0^\infty |\hat{u_k}(w)|^2 dw}. \tag{6}$$

VMD was applied to the experimental FBG signal, shown in Figure 4a, and the BLIMFs from the VMD of the experimental signal with the equal decomposition number of the EMD and the balance parameter set to 400 can be seen in Figure 4c. Figure 4b depicts the IMFs extracted from the EMD of the FBG signal. It is obvious that the intrinsic modes $u_6$ and $u_7$ contained little information that the VMD decomposition was overbinning, and the results were redundant. Thus, the VMD predefinition method where the basic mode number $K$ was equal to the EMD decomposition level was unreasonable, and a better use for the FBG experiment data can be seen in the following section.

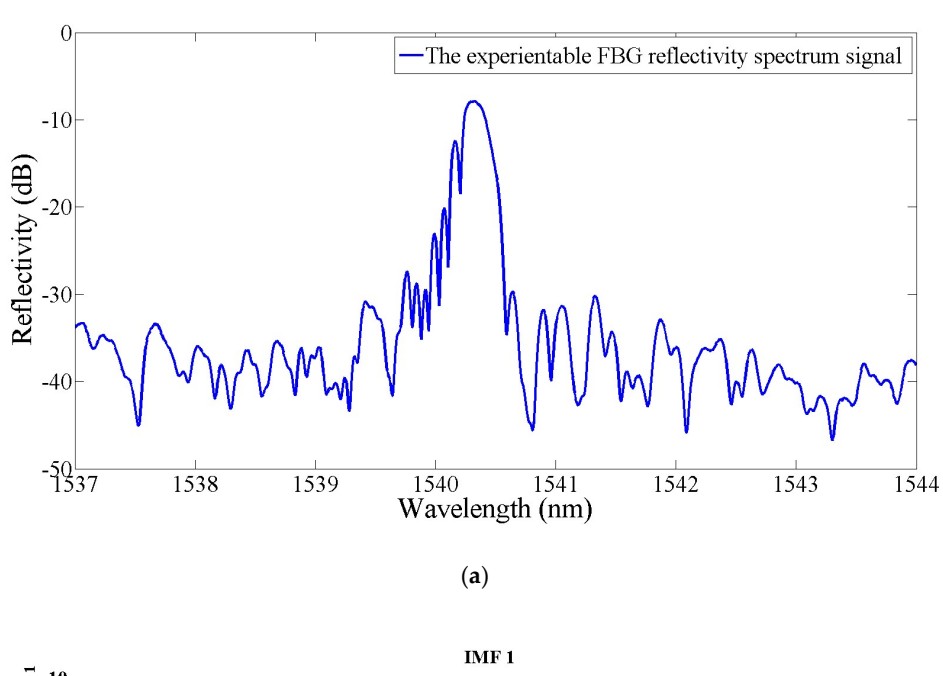

(a)

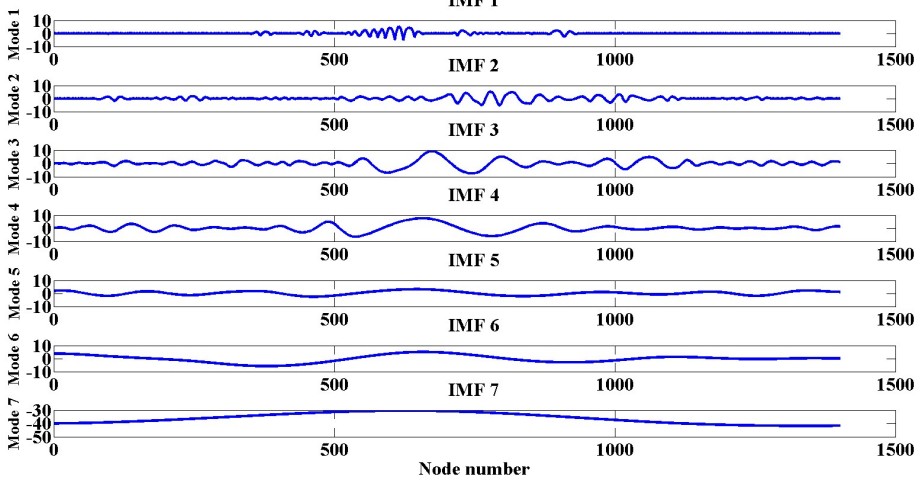

(b)

**Figure 4.** *Cont.*

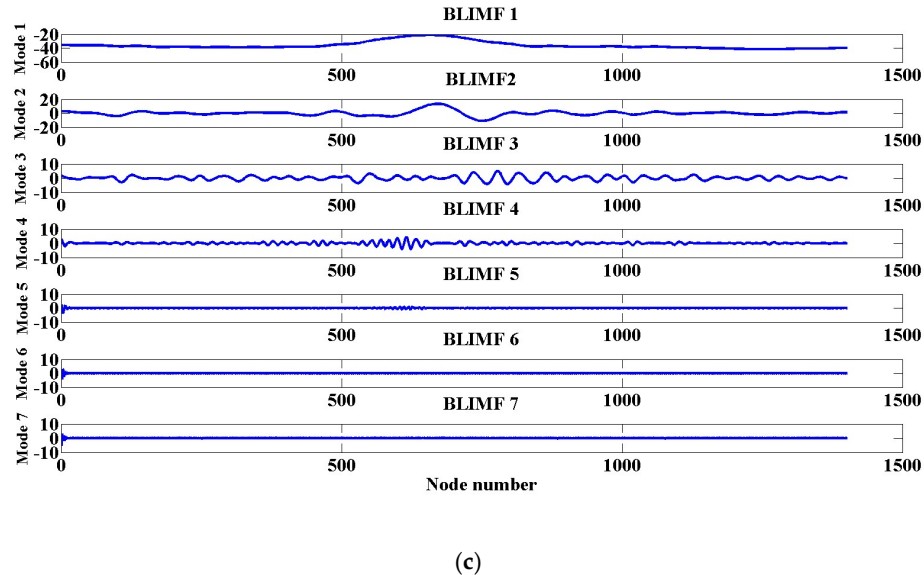

(**c**)

**Figure 4.** Decomposition of FBG reflectivity spectrum signals via EMD and VMD: (**a**) The FBG reflectivity spectrum signal, (**b**) the IMFs of the EMD via decomposition the reflectivity spectrum signal, and (**c**) the BLIMFs of the VMD via decomposition the reflectivity spectrum signal.

### 3.2. The Improved VMD-DWT Denoising Algorithm

The thresholding value could be self-adapted with spectral characteristics under different signal models resulting from VMD. An improved wavelet-VMD algorithm based on the standard wavelet-VMD is proposed for the spectrum signal denoising. Three key techniques were adopted in this section: one is wavelet thresholding, another is the data-fidelity constraint parameter, and the third is the decomposition number.

Compared with the traditional DWT denoising algorithm, the improved wavelet thresholding in the proposed algorithm considers the energy in each basic mode, and the denoising result is superior. Furthermore, in the traditional VMD algorithm, the decomposition number $K$ is required to be predefined, and its value has a predictable impact on the efficiency of filtering. In the literature, the basic mode number $K$ is usually defined with the same EMD decomposition number and the wavelet packet number or is selected by calculating the scaling exponent $\alpha$ of the input signal. However, the technologies above have a limitation in the experiment signal for the complicated frequency components. Therefore, the number $K$ was selected based on experience. In addition, the value of the data-fidelity constraint parameter $\alpha$ can be estimated by analyzing the anti-noise performance at different $\alpha$ values for the mixed signal of various $SNR_{input}$. The flow chart of the VMD-DWT denoising algorithm is shown in Figure 5.

The real FBG signal $f(i)$ with finite length $N$ is described as follows:

$$f(i) = s(i) + n(i) \quad i = 0, 1, 2, \ldots, N - 1 \tag{7}$$

where $s(i)$ is the FBG noise free signal, and $n(i)$ denotes random noise, which is associated with environment. The purpose of the denoising method is to find an estimated $\widetilde{f}(i)$ of the signal $f(i)$ with small error, i.e., completely remove the uncertain external disturbance noises from the experimental signals. In order to simplify the operation, we assumed that the $SNR_{out}$ denotes the signal/noise ratio (SNR) of the output signal and the $SNR_{input}$ indicates the SNR of the input signal. Based on the aforementioned analysis, the process of the VMD-DWT signal de-noising algorithm is shown as follows:

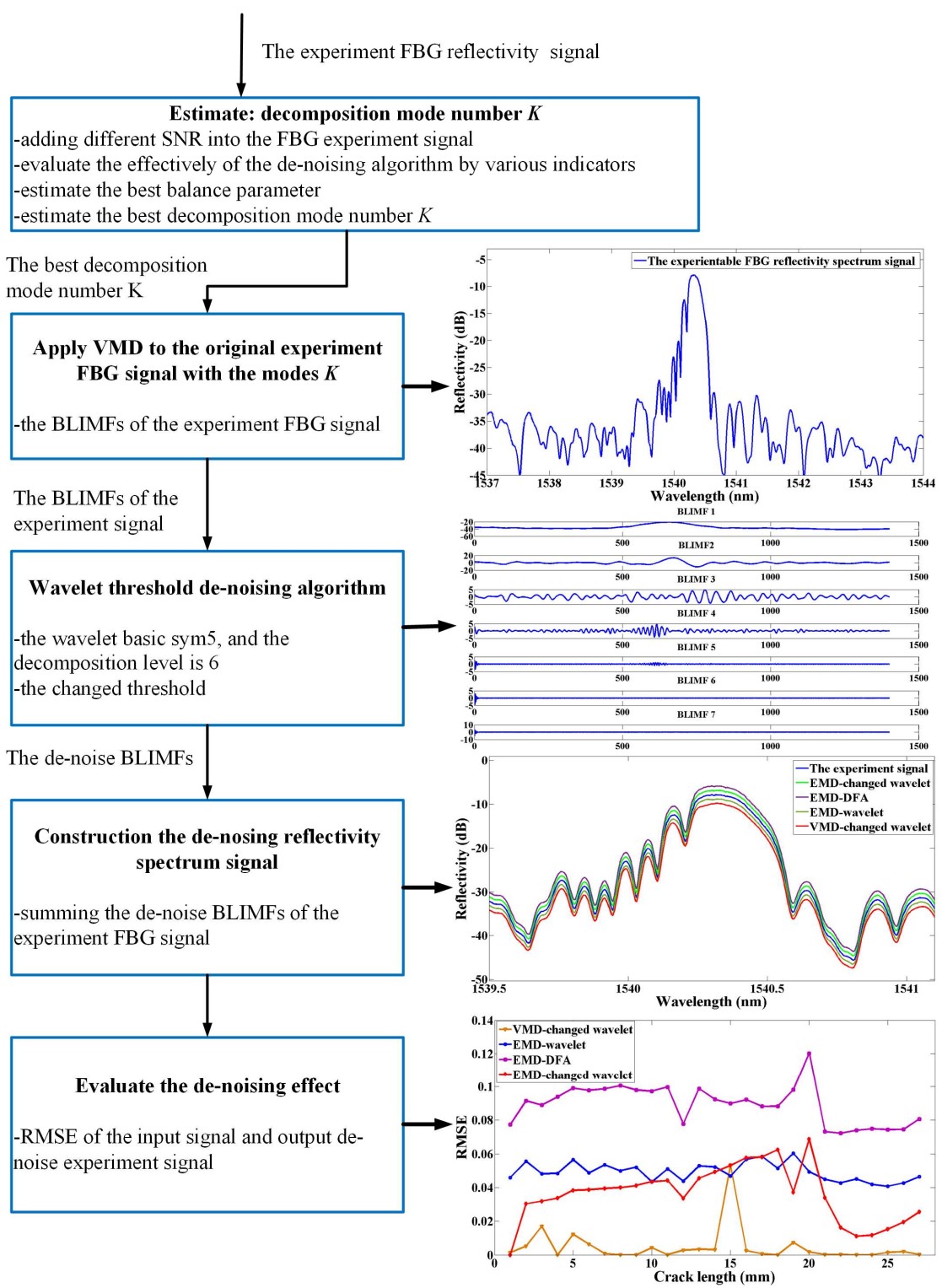

**Figure 5.** Flow chart of the wavelet-VMD based denoising algorithm.

Step 1: The decomposition number $K$ of the experiment signal is estimated. As discussed above, the decomposition number is not universally predefined. Thus, the number $K$ is defined based on experience. Considering the difficulties of calculating quantitative evaluation indexes, the mixed FBG signals by adding various $SNR_{input}$ Gaussian white noise to the experimental signals has been used in this paper to find the relationship between the decomposition number $K$ and the denoising results. In addition, the anti-noise performance of the proposed algorithm at different $\alpha$ values is analyzed

with mixed signals under various SNR$_{input}$ values. The best decomposition number $K$ and the balance parameter $\alpha$ for the realistic experiment signal are estimated based on the denoising results. Details can be found in Section 4, Case 1.

Step 2: The best decomposition number $K$ is determined. An input signal with length $N$ is decomposed into $K$ band-limited intrinsic mode functions (BLIMFs) $u_k$ by VMD, with respect to the sub-energy of the signal. The signal of length $N$ in the time domain is extended $2N$ to address the frequency domain.

Step 3: In wavelet decomposition, the predetermined and fixed equivalent filter-bank structure is not possible. A translate wavelet is performed by using thresholding in each BLIMF. Thus, the high-order BLIMF parts considered to be significantly corrupted by noise can be locally excluded. Based on a previous study, it is better to adopt DWT sym5 as the wavelet basis and to set the decomposition level at 6 [24].

This method is different from the traditional direct application of wavelet-like thresholding to the decomposition modes, hard or soft. In practice, the result of soft thresholding shows greater efficacy than the hard thresholding in FBG signals [24]. Soft thresholding is then applied to all BLIMF samples when the samples' extremums exceed the threshold, which means that they correspond to a zero-crossing interval and need to be reduced in a smooth way. Thus, the extremum is reduced by an amount exactly equal to the threshold. Considering the energy in each $u_k$, the improved wavelet thresholding is presented in Equation (8).

$$\hat{d}_l = f(x) = \begin{cases} sign(d_i)(|d_i| - \frac{T_i}{\exp(\frac{|d_i|}{T} - 1)^2}), & |d_i| \geq T_i \\ 0, & |d_i| \leq T_i \end{cases} \tag{8}$$

Here, $d_i$ presents the $i_{th}$ thresholding BLIMFs, and the standard deviation of the noise is estimated by a robust estimator leading to more accurate estimates, based on the components' median.

$$\sigma = \frac{median(|c_i| : i = 1, 2 \ldots . N)}{0.675}. \tag{9}$$

As previously mentioned, the noise contained in the BLIMF samples is not standard distributed with variance energy in each mode. In this sense, the reason for adapting a different thresholding $T_i$ per mode $i$ will become clear in the sequel for the scale dependence. Mathematically, the described soft thresholding operation yields Equation (9):

$$T = 100\sigma\sqrt{2E_i \lg(N)}/\lg(j+1). \tag{10}$$

Here, the $E_i$ shows the energy of the $K_i$ BLIMFs and can be computed directly based on the variation estimate of the first BLIMF using Equation (11).

$$\hat{E}_k = \frac{E_1^2}{\beta}\rho^{-k}, \ k = 2, 3, 4 \ldots \tag{11}$$

where $E_1^2$ is the energy of the first BLIMFs, and $\beta$ and $\rho$ are parameters for a specific VMD implementation, mainly depending on the number of shifting iterations used. They can be estimated based on the large number of independent noise realizations and their corresponding BLIMFs. In Reference [25], the parameters $\beta$ and $\rho$ are 0.719 and 2.01, respectively.

Step 4: After performing the synthesis function on each scale separately, finally, the de-noising signal is reconstructed by summing all processed BLIMFs after the denoising methods.

### 4. Results and Discussion

This section should provide a concise and precise description of the experimental results, their interpretation, and the experimental conclusions that can be drawn.

It is well known that the EMD-wavelet algorithm has been successfully used in removing the WGN from noisy signals. However, few studies on VMD-based denoising have been conducted. The EMD is highly dependent on methods of extremal point findings, which are sensitive to noise. The robustness of the algorithm may be reduced by the degrees of freedom. To overcome these limitations, a novel VMD decomposition scheme was used to obtain the recovered signal. The actual signal was processed to obtain the best decomposition number *K*. The real experimental signal was processed for a quantity evaluation of the denoising ability for four decomposition methods: the EMD-wavelet, EMD-DFA, EMD-wavelet-changed threshold, and VMD-DFA. The details are shown in Cases 1 and 2.

**Case 1:** Applying VMD to Mixed FBG Noisy Signals

As discussed before, the predefined decomposition number *K* is key to the VMD method. Previous studies set the number *K* to coincide with the EMD decomposition number. VMD may have an overbinning decomposition. The DFA method can also be applied to the mode number *K* definition. This may cause substantial simulation and experiment work since the FBG signal has not been studied before. In this paper, we propose an experimental way of solving the decomposition number definition problem. In Case 1, we obtained the actual FBG noisy signal by adding different SNR levels of white Gaussian noise to the experimental FBG signal. To evaluate the de-noising effect, quantitative evaluating indexes, such as the mean absolute error (MAE), the root-mean-square error (RMSE), and the SNR, are proposed here to assess the de-noising methods [26,27]. Additionally, the cross-correlation coefficient is proposed to estimate the correlation between the noisy signal and the denoising signal.

$$\text{MAE} = \frac{1}{N} \sum_{i=1}^{N} \left( s(i) - \widetilde{f}(i) \right)^2 \tag{12}$$

$$\text{RMSE} = \sqrt{\frac{1}{N} \sum_{i=1}^{N} \left( s(i) - \widetilde{f}(i) \right)^2} \tag{13}$$

$$\text{SNR} = 10\lg \left[ \frac{\sum_{i=1}^{N} s^2(i)}{\sum_{i=1}^{N} \left( s(i) - \widetilde{f}(i) \right)^2} \right] \tag{14}$$

$$\text{Coss correlation} = \frac{Cov\left( s(i), \widetilde{f}(i) \right)}{\sqrt{Var(s(i))\,Var\left( \widetilde{f}(i) \right)}} \tag{15}$$

where $\widetilde{f}(i)$ represents the de-noising FBG signal, and the $s(i)$ values are the noisy FBG signals. $Cov(*)$ indicates the covariance function, and $Var(*)$ shows the variance equation.

The value of the data-fidelity constraint balance parameter $\alpha$ is inversely proportional to the bandwidth of the signal mode and closely related to anti-noise performance. The bandwidth mode widens as $\alpha$ decreases, and the modes may have poor anti-noise performance. However, the center frequency mode will be less accurate according to the bandwidth mode, which decreases with a greater $\alpha$. Based on a previous study [3], the veracity of the damage of the primary wavelength relies heavily on the center frequency of extracted modes. Therefore, it is necessary to find a proper value of $\alpha$, by not only considering the center frequency accuracy but also taking the anti-noise performance into account.

In order to find a balance point of $\alpha$, the anti-noise performance under different $\alpha$ values was analyzed with different $\text{SNR}_{\text{input}}$ mixed signals. The central frequency of Mode 1 corresponds to the

central wavelength considered the damage characteristic, and the primary wavelength shifting with different crack lengths was calculated with various values of $\alpha$. The decomposition level was set as 6. The interval of $\alpha$ was set within [1, 5000] according to research indicating that the mode separation error is intolerable when $\alpha$ is over 5000 [28]. Hence, under different $\alpha$ values that increase from 1 to 5000 with a step width equal to 100, the center frequency of Mode 1 was calculated sequentially, and the result can be seen in Figure 6. It was found that with a larger $SNR_{input}$, the center frequency of noisy signals approached the center frequency of original signals more closely, which means a better anti-noise performance. It is worth noting that the central frequency of Mode 1 in different SNR noise signals most closely approaches the central frequency of the original signal with good anti-noise performance when $\alpha$ is approximately 200. In this research, the value of $\alpha$ was set to 200 to guarantee the value accuracy of the center frequency when the experiment signal was processed later.

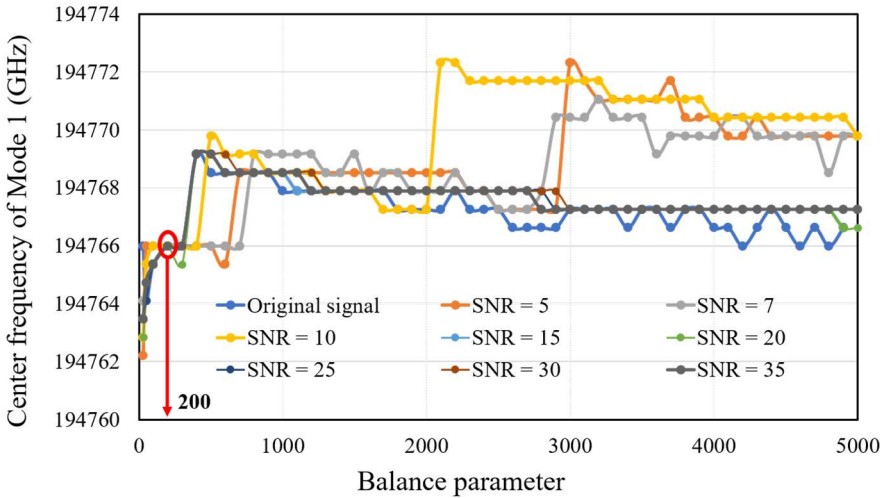

**Figure 6.** $\alpha$ of Mode 1 with mixed and original signals under various SNRs.

Taking the mixed FBG noisy signals shown in Figure 4a as an example, the proposed denoising algorithm was applied to the samplings, and the results are given in Table 1. The EMD-wavelet, EMD-DFA, and EMD-changed thresholding wavelet (named EMD-DWT) algorithms are used for comparison due to their well-developed characteristics in terms of removing WGN from signals. The parameters of VMD were also chosen as follows: $\alpha = 200$, $\tau = 0$, and $\varepsilon = 1 \times 10^{-7}$. EMD with the traditional wavelet thresholding method was chosen as the basic denoising algorithm to evaluate the effectiveness of other three methods. Moreover, the EMD-DWT algorithm was used to assess the denoising performance under the VMD and EMD domains. Moreover, the traditional denoising method of EMD-DFA was proposed as a reference algorithm. The mixed FBG noise signal with SNRs from 5 to 35 dB were processed, and the quantity evaluation results can be seen in the following table. Table 1(a) presents the $SNR_{out}$ of the above four denoising algorithms, Table 1(b) presents the cross correlation, and Table 1(c) presents the MSE of the above-mentioned denoising methods. The bold numbers in the figure below presents the best $SNR_{out}$ of the three traditional algorithms and the proposed algorithm under different decomposition numbers.

**Table 1.** (**a**) Comparisons the $SNR_{out}$ of the VMD with changed thresholding wavelet and EMD denoising methods for mixed FBG signal. (**b**) Comparisons the cross correlation of the VMD with changed thresholding wavelet and EMD denoising methods for mixed FBG signal. (**c**) Comparisons the MAE of the VMD with changed thresholding wavelet and EMD denoising methods for mixed FBG signals.

| (a) | | | | | | | | |
|---|---|---|---|---|---|---|---|---|
| SNR (dB) | 5 | 7 | 10 | 15 | 20 | 25 | 30 | 35 |
| EMD-wavelet (soft) | **28.27289** | **29.03794** | **29.94879** | **34.62247** | 36.03779 | 38.70505 | 40.36963 | 40.66328 |
| EMD-DFA | 21.45639 | 20.79891 | 27.80233 | 20.84781 | 37.91600 | 43.04961 | 48.30413 | 53.24952 |
| EMD-DWT | 22.98264 | 24.93801 | 27.80229 | 33.08285 | **37.91601** | **43.04964** | **48.30413** | **53.24952** |
| VMD-DWT | | | | | | | | |
| $K = 1$ | 18.79288 | 18.65518 | 18.38846 | 18.80969 | 26.82166 | 27.21651 | 32.46437 | 32.66283 |
| $K = 2$ | 23.01841 | 23.64160 | 19.52073 | 24.16615 | 29.21863 | 32.06180 | 34.01615 | 39.82442 |
| $K = 3$ | 25.09874 | 25.09611 | 22.54665 | 27.92866 | 30.06698 | 37.79198 | 37.94835 | 44.41590 |
| $K = 4$ | **28.44365** | **29.81668** | 24.83959 | 29.63455 | 33.04874 | 38.72096 | 39.84002 | 47.23453 |
| $K = 5$ | 25.99756 | 25.96998 | **29.97101** | **34.78591** | **37.98547** | **43.04964** | 45.16942 | 50.30090 |
| $K = 6$ | 25.34463 | 25.96834 | 28.86079 | 31.42557 | 35.30840 | 39.04831 | **48.58089** | **53.24953** |
| $K = 7$ | 24.86184 | 25.77156 | 24.81353 | 31.26708 | 34.64873 | 37.34640 | 32.33718 | 47.70035 |
| $K = 8$ | 24.32871 | 25.3521 | 24.18736 | 30.30249 | 33.72422 | 37.22826 | 30.94072 | 44.94072 |
| $K = 9$ | 24.12876 | 24.8201 | 23.26579 | 30.10474 | 32.6836 | 36.1196 | 29.35118 | 41.35118 |
| $K = 10$ | 24.09218 | 24.5655 | 22.57683 | 29.42232 | 32.57326 | 35.2429 | 28.75985 | 38.75985 |

| (b) | | | | | | | | |
|---|---|---|---|---|---|---|---|---|
| SNR (dB) | 5 | 7 | 10 | 15 | 20 | 25 | 30 | 35 |
| EMD-wavelet (soft) | **0.999211** | **0.999341** | **0.999463** | **0.999816** | 0.999867 | 0.999928 | 0.999951 | 0.999954 |
| EMD-DFA | 0.996210 | 0.995574 | 0.999120 | 0.995617 | 0.999914 | 0.999974 | 0.999992 | 0.999997 |
| EMD-DWT | 0.997322 | 0.998289 | 0.999120 | 0.999738 | **0.999914** | **0.999974** | **0.999992** | **0.999997** |
| VMD-DWT | | | | | | | | |
| $K = 1$ | 0.993307 | 0.993061 | 0.999120 | 0.993247 | 0.997264 | 0.996479 | 0.989061 | 0.999711 |
| $K = 2$ | 0.997981 | 0.997784 | 0.983889 | 0.998011 | 0.998034 | 0.997344 | 0.999127 | 0.999763 |
| $K = 3$ | 0.998470 | 0.998468 | 0.998160 | 0.997895 | 0.999763 | 0.998193 | 0.999370 | 0.999816 |
| $K = 4$ | **0.998510** | **0.999638** | 0.998997 | 0.999097 | 0.999880 | 0.999732 | 0.999457 | 0.999899 |
| $K = 5$ | 0.998343 | 0.999466 | **0.999497** | **0.999897** | **0.999922** | **0.999974** | 0.999594 | 0.999937 |
| $K = 6$ | 0.998452 | 0.998621 | 0.999261 | 0.999794 | 0.999865 | 0.999819 | **0.999921** | **0.999997** |
| $K = 7$ | 0.998267 | 0.998529 | 0.998792 | 0.999702 | 0.999719 | 0.999374 | 0.999250 | 0.999891 |
| $K = 8$ | 0.998037 | 0.998475 | 0.998582 | 0.999699 | 0.999638 | 0.998787 | 0.998633 | 0.999633 |
| $K = 9$ | 0.997944 | 0.998361 | 0.997440 | 0.999618 | 0.999506 | 0.995454 | 0.998395 | 0.999395 |
| $K = 10$ | 0.997924 | 0.998315 | 0.997741 | 0.999606 | 0.999426 | 0.992086 | 0.998349 | 0.999349 |

| (c) | | | | | | | | |
|---|---|---|---|---|---|---|---|---|
| SNR (dB) | 5 | 7 | 10 | 15 | 20 | 25 | 30 | 35 |
| EMD-wavelet (soft) | **0.241370** | **0.213610** | **0.177919** | **0.125300** | 0.083814 | 0.063342 | 0.053885 | 0.052942 |
| EMD-DFA | 0.389650 | 0.364545 | 0.254550 | 0.323424 | 0.079609 | 0.044027 | 0.024160 | 0.013852 |
| EMD-DWT | 0.439771 | 0.354689 | 0.254553 | 0.139824 | **0.079609** | **0.044027** | **0.024160** | **0.013852** |
| VMD-DWT | | | | | | | | |
| $K = 1$ | 0.640044 | 0.653047 | 0.654553 | 0.633102 | 0.633446 | 0.622933 | 0.620225 | 0.518134 |
| $K = 2$ | 0.315766 | 0.321990 | 0.448884 | 0.279824 | 0.276412 | 0.317596 | 0.234400 | 0.217975 |
| $K = 3$ | 0.369700 | 0.296232 | 0.340961 | 0.291351 | 0.186823 | 0.117153 | 0.113686 | 0.114086 |
| $K = 4$ | **0.296320** | **0.228715** | 0.265572 | 0.172845 | 0.157558 | 0.055798 | 0.073030 | 0.087924 |
| $K = 5$ | 0.332780 | 0.261142 | **0.17698** | **0.119602** | **0.079603** | **0.044027** | 0.052551 | 0.042896 |
| $K = 6$ | 0.317743 | 0.297420 | 0.259824 | 0.138284 | 0.103971 | 0.085524 | **0.022050** | **0.013030** |
| $K = 7$ | 0.349587 | 0.305211 | 0.295239 | 0.142640 | 0.114192 | 0.107896 | 0.094784 | 0.076756 |
| $K = 8$ | 0.369039 | 0.314728 | 0.339887 | 0.145079 | 0.123990 | 0.119463 | 0.106430 | 0.086430 |
| $K = 9$ | 0.381169 | 0.318587 | 0.470725 | 0.149699 | 0.137336 | 0.121081 | 0.118543 | 0.078543 |
| $K = 10$ | 0.384062 | 0.325191 | 0.427981 | 0.156916 | 0.138992 | 0.132615 | 0.129593 | 0.069593 |

For the VMD-changed thresholding wavelet algorithm, the decomposition number $K$ determines the proposed algorithm denoising performance. According to previous research, the results of three parameters, the $SNR_{out}$, the MAE, and the correlation coefficient, to quantitatively evaluate the

denoising performance, were used to analyze the best VMD decomposition number under different $SNR_{input}$ values. It is clear that when the decomposition number *K* was set too high, too many basic modes were obtained (overbinning); when the decomposition number was set too low, there may be too few basic modes (underbinning). The three quantitative evaluating parameters show that the denoising effect apparently does not reach the best expectations. Additionally, *K* is related to the SNR of the input signal. A higher $SNR_{input}$ of mixed signals corresponds to a higher decomposition level. The best decomposition level number is 4 with a mixed signal with a low $SNR_{input}$; however, with a mixed signal with a high $SNR_{input}$, the best basic mode number increases and maintains stability at 6. On balance, for a realistic experimental signal without adding noise, the best decomposition level should be 6 in the denoising process.

In Table 1, the EMD-changed thresholding wavelet algorithm had better performance than the traditional EMD-wavelet denoising method, which illustrates that the changed thresholding was efficient at signal denoising. Moreover, comparing other algorithms, such as the EMD-wavelet method, the EMD-DWT algorithm, and the EMD-DFA method, the changed thresholding VMD-DWT algorithm showed the highest SNR and cross-correlation with the smallest MAE in four algorithms when the decomposed number and balance parameter were chosen optimally. The quantitative denoising analysis results indicate that the proposed algorithm had the best denoising effect, and among all four methods mentioned, was the most robust. Additionally, the changed thresholding VMD-DWT algorithm effectively removed noise from experimental signals, laying the foundation for further experiments in FBG signal denoising operations.

**Case 2:** The Experimental FBG Signal

In Case 1, the best decomposition number level *K,* equaling 6, and the balance parameter $\alpha$, equaling 200, were obtained to process the realistic experimental signals collected from the damage monitoring experiment mentioned above. The EMD-wavelet algorithm, EMD-DFA denoising method, and the EMD-DWT method were selected for comparison to evaluate the denoising ability of the proposed algorithm. The intuitive denoising analysis results of the initial experimental signals can be seen in Figure 7a, and the noise that eliminated the realistic experiment signal is clearly shown in Figure 7b. It can be seen that the VMD-changed thresholding wavelet could remove the noise more completely than the other three algorithms.

To evaluate the proposed algorithm and other algorithms, the RMSE was proposed as a quantitative evaluation index based on the literature [13]. The RMSE results of different denoising algorithms can be seen in Figure 8. The figure shows the results of the RMSE indicator being used to quantity the denoising ability of the proposed algorithm. For a quantitative index, the standard deviation $\sigma$ between the initial realistic experimental signal and the denoising signal was calculated, and is mathematically expressed as in Equation (2).

The RMSE results presented at different crack lengths during the damage monitoring experience are shown in Figure 8. The changed thresholding VMD-DWT denoising algorithm has the lowest RMSE compared with the other methods. Thus, the effectiveness of this proposed algorithm in denoising experimental FBG signals was verified. This technique has thus been successfully validated in experimental conditions. These results may be used in crack growth monitoring.

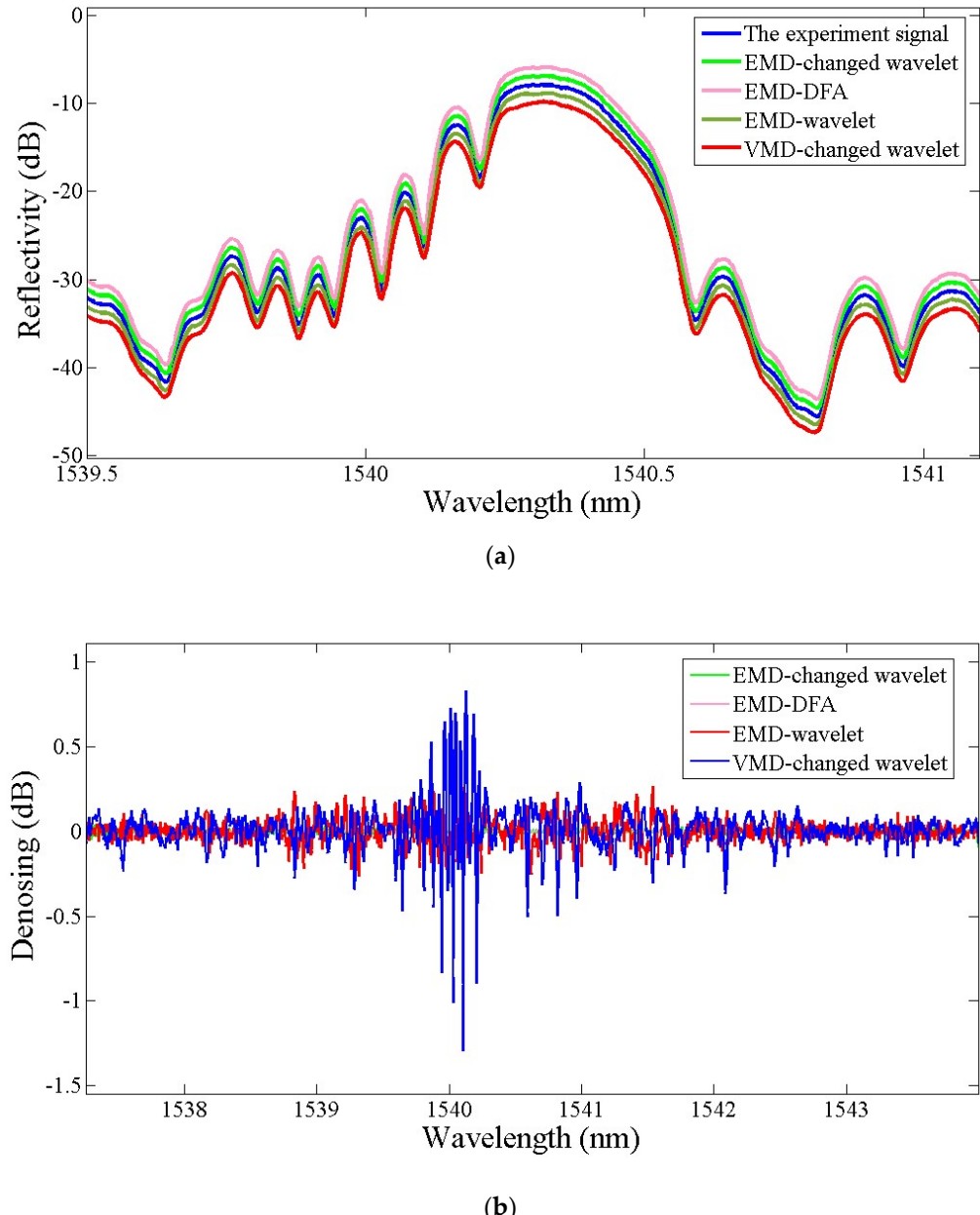

(**a**)

(**b**)

**Figure 7.** (**a**) VMD-changed thresholding denoising of the FBG experimental signal. The original spectrum (blue) is filtered through a VMD→DWT→changed thresholding→IDWT→IVMD. (**b**) The noisy part of the experiment FBG signal after denoising.

In order to denoise the FBG signals obtained from the fatigue crack detection experiment, we proposed a VMD combined with the changed thresholding wavelet denoising method. The thresholding definition in the algorithm considers the energy distribution in each basic mode. Furthermore, in order to deal with the best balance parameters $\alpha$, which was obtained by comparing the central frequency of the original and mixed signals with well overlapped at different parameter numbers. Different to other definition methods, the central frequency corresponds to the physical parameter central wavelength of the spectrum. Additionally, the denoising performance under different $\text{SNR}_{\text{input}}$ mixed signals are used to acquire the best decomposition number *K*. It is obvious that the proposed algorithm showed the best denoising ability for mixed signals for the highest $\text{SNR}_{\text{input}}$ and cross correlation, compared with the traditional signal denoising methods, such as the EMD-wavelet (soft), EMD-DFA, and the EMD-DWT algorithm. Furthermore, in the fatigue crack

propagation detection experiment, the real FBG signals were processed by four denoising methods, and the results show that the proposed algorithm was superior than the other traditional methods, since it retained the useful damage signal information and removed environment noise. Therefore, the definition methods of the best decomposition number and balance parameter can be used in the VMD-changed wavelet methods and the proposed denoising methods can process the FBG signal for fatigue crack detection.

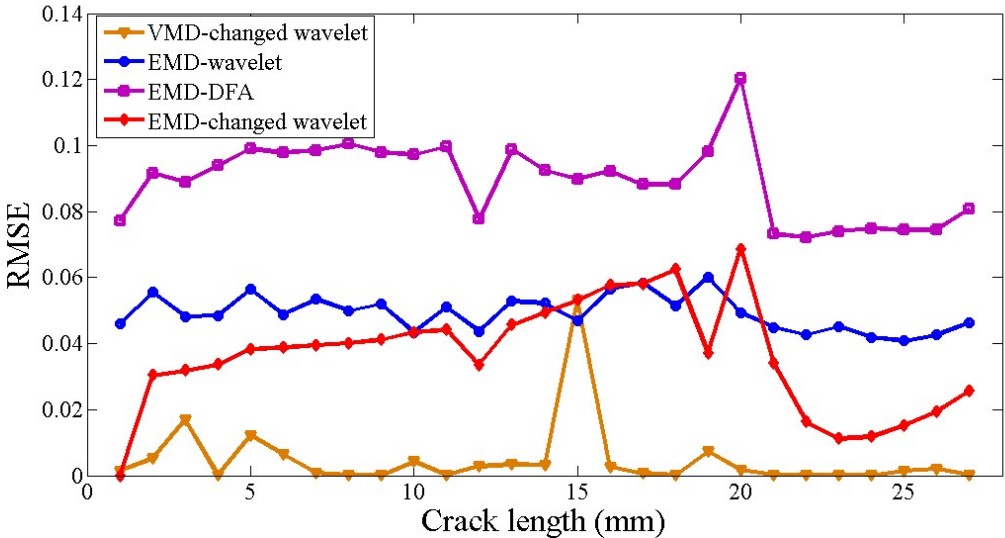

**Figure 8.** The RMSE of four signal denoising algorithms at different crack lengths.

## 5. Conclusions

In this paper, a recently introduced VMD was proposed to filter the mixed and real noisy signals contaminated by FBGs. The main problem in VMD is selecting the decomposition number *K* of the basic modes of BLIMFs and the balance parameter $\alpha$. To solve the problem, mixed FBG signals were used to find the best decomposition number *K* and the data-fidelity constraint $\alpha$ for VMD denoising, and $SNR_{out}$, the MAE, and the correlation coefficient were presented as denoising performance indicators. In Case 1, the best balance parameter $\alpha$ was obtained by analyzing the anti-noise performance at different $\alpha$ values. The mixed FBG input signals with different SNR noises were broken down into the given numbers *K* of BLIMFs through VMD. After applying the changed thresholding wavelet to the BLIMFs, the filtered signals were reconstructed from modes by summing the de-noised BLIMFs. Additionally, the previously mentioned three indexes were used to evaluate the denoising performance. The best decomposition number *K* could be defined when the VMD-changed thresholding wavelet algorithm had the best denoising effect. It was found that the denoising results of the proposed algorithm for the mixed signals were different. Furthermore, considering the mixed FBG signals, the denoising results show that the proposed algorithm is superior to the EMD-changed thresholding wavelet, the EMD-DFA, and the EMD-DWT filtering techniques. To illustrate the effectiveness of this proposed method, we carried out Case 2. The input signal data were collected from the crack monitoring experiment described in Section 2. Based on Case 1, the best decomposition number *K* was set at 6, and the balance parameter was set to 200. It was found in the quantitative results of RMSE indicators that the capability of VMD-changed wavelet methods in FBG signal denoising was higher than that of other denoising algorithms. As previous discussed, the parameter of the proposed definition methods considers the FBG signal physical characteristics and the VMD-changed thresholding wavelet algorithm denoising performed better than other three traditional methods, which can be used in FBG signals processing in the SHM field.

**Author Contributions:** M.Z. and W.D. conceived the key idea and designed the experiments, W.Z. provided the academic support and checked the manuscript, and Y.Z. and B.J. performed the experiment. All authors made contributions to the writing and revising of the manuscript.

**Funding:** This work was supported by the Natural foundation program (grant number 51705015) from the National Natural Science Foundation of China, and the Technical foundation program (grant JSZL2017601C002) from the Ministry of Industry and Information Technology of China.

**Conflicts of Interest:** The authors declare no conflict of interest.

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
