# Peer review of "Denoising of the Fiber Bragg Grating Deformation Spectrum Signal Using Variational Mode Decomposition Combined with Wavelet Thresholding"

_applsci, doi:10.3390/app9010180_

Round 1

Reviewer 1 Report

Damage detection using FBG sensors is proposed in this contribution. Due to low SNR of captured signal, appropriate signal processing is required, primarily related to signal denoising. As the VMD alongside changed-thresholding is used here, the reviewer cannot agree that the method is "novel method" (both of them are already applied). However, comparison of different methods is worthy.

The novelty has to be discussed in more clear way and pointed out as it seems now that only already applied methods are applied and compared.

State-of-Art (introduction) has to be organized in such a way that the difference between already published methods and proposed one in considered contribution becomes clear.

Font size in Figure 4 and Figure 5 is too small (not readable). It has to be changed. Also the font is not unified in complete contribution (for example: Figure 6 - bold labels).

Please check the guidelines once again ... for example: "3.1. the imp" should be "3.1. The imp".

Extensive editing of English language is necessary.

Author Response

Thank you very much for your useful comments and suggestions on the content and language of our manuscript. Your comments are very valuable and helpful for revising and improving our work. We have seriously considered each comment and modified the manuscript accordingly. The detailed corrections are seen in the annex.

Reviewer 2 Report

This article deals with a signal denoising method which combines VMD and changed-thresholding wavelet to denoise the experimental and mixed signals. The main problem in VMD is to select the decomposition number of the basic modes about BLIMFs and the balance parameter. To solve the problem, the mixed signals which are obtained by adding different SNR noises to the experimental signals are designed for damage detection of FBG sensor. Overall, the paper has a good structure and the technical quality is fair. In my opinion, the paper is well presented, but I have found minor issues as follows: (i) In Introduction, in order to clarify the purpose and scope, and main issue of the article, the main contribution of the authors should discuss more profoundly and highlights in detail. (ii) The consistency of Fig. 5 and 6 are not sufficient, and thus graphic scheme of these figures should be given clearly. (iii) In this article, the uses of some symbols are confusing. Therefore, a nomenclature section should be added to help the reader understand and follow better the variables used throughout the paper.

Author Response

(The authors gave the same response as above.)

Round 2

Reviewer 1 Report

1. In Figure 4b and 4c there are still no labels. It should be changed. Other comments are taken in consideration.

Author Response

(The authors gave the same response as above.)
